# Mask - Region-based Convolutional Neural Networks (R-CNN) with Radiomics Integration and Gray Level Co-occurrence Matrix (GLCM) for brain tumor detection and segmentation

**Prathima Devadas**[1]*, **Gandhi Mathivanan**[2]

**1** Department of Computer science and Engineering, Sathyabama Institute of Science and Technology, Chennai, India, **2** Department of Computer science and Engineering, Sathyabama Institute of Science and Technology, Chennai, India

* prathima.dev20@outlook.com

## Abstract

Early diagnosis of brain tumors is important for successful treatment and better patient consequences in industrial information systems. This research employs Mask Region-based Convolutional Neural Networks (R-CNN), radiomics integration, and the Gray Level Co-occurrence Matrix (GLCM) to progress brain tumor detection and segmentation using Magnetic Resonance Imaging (MRI) images. The Mask R-CNN is employed to precisely label tumor regions of interest (ROIs) by precisely segmenting pixels. Radiomic features in the form of texture, shape, and intensity measures are calculated on the segmented ROIs to provide a numerical estimate of tumor heterogeneity. Meanwhile, GLCM-based texture features are computed to derive the fine-scale spatial classifications of pixel intensities in tumor regions, which may provide more information regarding tumor structure. Such handcrafted features are merged with deep features obtained with the help of the Mask R-CNN architecture to form a strong feature set that takes the benefits of both classical radiomics and deep learning (DL). The combination of the feature set is classified using a detection head multilayer perceptron (MLP) to predict the presence and the nature of a tumor. The experimental results on an experimental brain tumor MRI dataset proved that the proposed approach is better than independent Mask R-CNN and radiomics-based approaches. The model is very sensitive, specific and precise in differentiating between glioma, meningioma, pituitary adenoma and non-tumor cases. The feasibility of this hybrid solution is the combination of handcrafted and deep features to further the brain tumor detection that provides medical practitioners with a handy tool of detecting brain tumors early and accurately.

**Data availability statement:** All relevant data are within the manuscript.

**Funding:** The author(s) received no specific funding for this work.

**Competing interests:** The authors have declared that no competing interests exist.

## Introduction

The brain is one of the most complex organs in the human body, consisting of billions of cells. This structure is essential to the neurological system's operation. It evaluates sensory data, controls physiological processes, and interacts with the body by sending the appropriate signals after deciding on the optimal course of action [1]. Brain tumors, which are brought on by abnormal cell growth in brain tissues, are among the most fatal forms of brain disease. Primary and secondary brain tumors are the two forms of brain tumors. Primary brain tumors, which grow and stay contained within the brain, account for around 70% of all brain tumor occurrences. In contrast, secondary brain cancers originate in the breast, kidneys, or lungs before spreading to the brain. Approximately 200,000 people worldwide receive a primary or secondary brain tumor diagnosis each year [2]. Furthermore, there are two types of brain tumors: benign and malignant. For benign tumors, surgery is frequently a good treatment option. Malignant tumors, regrettably, are among the most deadly forms of cancer, with swift and severe outcomes. Although the precise source of brain tumors is still unknown, they can afflict people of any age, including children. Headaches, nausea, hand tremors, vision abnormalities, and behavioral or personality changes are all signs of brain tumors [3].

Brain cancers are often divided into three main categories: pituitary tumors, gliomas, and meningiomas [4]. Gliomas are the most prevalent of them. Higher grades of these big tumors, which develop in the brain's glial cells, raise the risk of developing cancer. The meninges, a protective layer of tissue, give rise to meningiomas, which are comparatively rare and usually benign. The pituitary gland is where pituitary tumors start. These tumors are generally thought to be benign and are less prevalent.

Early detection of brain tumors is vital for successful therapy. Advances in medical imaging have made imaging methods crucial for both diagnosing and treating diseases. Brain cancer can sometimes be detected with the assistance of Computer-Aided Diagnosis (CAD) methods using MRI techniques. CAD systems are critical software tools that assist radiologists in analyzing and interpreting MRI data quickly [5]. By providing precise information about a tumor's location, type, and size, imaging modalities help medical professionals make an accurate diagnosis of brain tumors. Radiologists can use this information to plan patient care effectively. MRI imaging is essential for detecting brain cancers because it uses electromagnetic influences and a strong magnetic field to differentiate among various anatomical elements. There is no need for multiplanar, multi-slice, or positional changes in MRI to obtain axial, parallel, and sagittal images of patients. Compared to Computed Tomography (CT) scans, MRI exposes patients to less radiation. Moreover, MRI offers better contrast features than CT. Because MRI can provide important details regarding the location, size, and shape of tissue structures, it has recently outperformed other techniques in detecting brain malignancies. MRI is frequently used to identify brain tumors and other illnesses like dementia, Parkinson's disease, and Alzheimer's [6–8].

## Research of our work

The Mask R-CNN has a complex radiomics and GLCM characteristics of the brain tumor detector. Mask R-CNN is a powerful DL semantic segmentation framework, which identifies and localizes tumor regions on MRI images on a pixel-scale. The quantitative features of these segmented regions are the radiomics characteristics of tumor texture, tumor shape and tumor intensity. One of them is called GLCM and is statistical technique used to study the spatial relation between pixel values forming important texture patterns giving information on tumor heterogeneity. The radiomics characteristics are then assigned to the appropriate categories of benign, malignant or normal brain tumors using a detection head (MLP) which is a sub-type of feed forward artificial neural network. Such a combination method takes advantage of the characteristic of segmentation of Mask R-CNN and interpretability of radiomics and high classification ability of MLP, which can offer a single model of reliable and automatic brain tumors margin.

## The contribution of our work are below:

- To enhance a brain tumor detection and segmentation method using MRI images through the assistance of the Mask R-CNN, radiomics and GLCM. The use of Mask R-CNN in pixel-by-pixel segmentation will result in accurate delineation of tumor ROIs.

- The radiomics characteristics (texture, shape and intensity feature) are derived on the basis of the segmented ROIs to give a quantitative evaluation of the tumor heterogeneity. Meanwhile, GLCM-based texture characteristics are approximated to provide more finer-grained spatial relations of pixel-brightness of tumor patches, which provide more data regarding tumor structure.

- These handcrafted features are then fused with deep features generated with the help of the Mask R-CNN architecture to generate a strong set of features that unites the advantages of classical radiomics and deep learning.

## Literature Survey

Convolutional neural networks (CNNs), and U-Net models in particular, have been widely used in the literature for brain tumor identification to accurately segment and organize tumors in MRI data. Techniques such as Class-Conditional Generative Adversarial Networks (GANs) have also been explored to enhance image quality and augment datasets. Studies have shown that hybrid models incorporating attention mechanisms, like Swin Transformers, further improve detection accuracy by focusing on key features. These methods aim to provide faster, more reliable diagnostic tools, assisting clinicians in early tumor detection and treatment planning.

To identify the cancers, Sharma et al. [9] introduced a distinctive model with the usage of neural features of MRI images depending on the histogram of gradients (HOG). More intuitive features were extracted using the feature optimization technique applied in this research to analyze the complex feature vector. We used the HOG methodology to improve a modified ResNet50. By employing deep learning techniques, the updated ResNet50 model is able to accurately extract deep features. To maintain optimal computing performance, this model is used in conjunction with the updated layered architecture. Additionally, we incorporated feature extraction and augmentation techniques with a machine learning-based ensemble classifier, which further delivers the optimal fusion vector for tumor recognition. With HOG and the modified ResNet50, this hybrid method establishes exceptional performance with an 88% detection accuracy.

To improve diagnostic accuracy, Alshuhail et al. [10] suggested a CNN-based DL model. The model utilizes a sequential CNN architecture with dropout, max-pooling, and several convolutional layers. Dense layers are then added for final classification. The suggested method significantly enhances diagnostic precision, attaining an overall test accuracy of 98%. Grad-CAM visualizations also enhance interpretability by showing how the model justifies its decision. The two main

 

issues that are tackled in this study are the different morphologies of the tumor and the fact that a quick and accurate diagnostic method is required in the classification of brain cancer using MRI.

Akter et al. [11] suggested a U-Net-based segmentation model and a deep CNN-based architecture to obtain the automatic classification of brain images into four classes. We tested the effects of segmentation on brain MRI tumor classification through the training of segmentation and testing of classification models on six benchmarked datasets. Moreover, we have evaluated two categorization algorithms using Area Under the Curve (AUC), accuracy, recall, and precision. The better of our deep learning-based model compared to the existing pre-trained models in brain tumor classification and segmentation is better in all six datasets. The results indicate that our classification model with the segmentation strategy was the most accurate in the combined dataset with 98.8 and 98.7 detection accuracy.

Rasheed et al. [12] proposed a novel way to classify primary brain tumors, such as glioma, meningoma, pituitary tumor, as well as non-tumor cases, through the integration of CNNs and a hybrid attention mechanism. Using benchmark datasets from reliable sources, the method was thoroughly assessed and associated with several well-known pre-trained models, such as DenseNet201, ResNet101V2, Xception, ResNet50V2, and DenseNet169. The experimental findings showed exceptional performance, with an F1-score of 98.20%, an overall detection accuracy of 98.33%, and precision and recall values of 98.30%.

Saurav et al. [13] proposed a new lightweight Attention-Guided Convolutional Neural Network (AG-CNN) to classify brain cancers in magnetic resonance imaging (MRI). The design concentrates on the most relevant regions of the image to classify tumors in an accurate manner using channel-attention blocks. The AG-CNN effectively merges the multi-level feature representations through the addition of skip connections and global average pooling, to enhance the extraction of discriminative features that are needed to differentiate the tissues of the brain that have tumors and those that are healthy. The model performed better than the advanced methods in terms of computational efficiency and resiliency on four benchmark MRI brain tumor datasets. The lightweight nature of the proposed AG-CNN can be easily implemented on embedded systems, and the classification of brain tumors in real-time can be easily applied in clinical environments with limited processing capabilities.

Sadr et al. [14] introduced a DL-based approach to the automatic detection of brain cancer using MRI images. They developed two models: one that was binary (normal vs. abnormal) and another that was multiclass. The models were trained using a total of 3,064 MRI images (Figshare dataset), 3,000 (Br35H) and 152 (Harvard Medical dataset). The Figshare dataset was primarily used to train a 26-layer deep CNN model due to its larger sample size. Transfer learning techniques were applied using the Br35H and Harvard Medical datasets to integrate optimized VGG16 and Xception architectures, thereby reducing overfitting in the proposed CNN. Experimental results on the Figshare dataset showed that the proposed Deep CNN achieved a detection accuracy of 97.27%.

Mahamud et al. [15] created CNNs and models based on transfer learning to classify three major types of brain malignancies: gliomas, meningiomas, and pituitary tumors. The data comprised 7,023 human brain MRI images, which were separated into 4 groups: glioma, meningioma, pituitary tumor, and no tumor. The results of the study were very precise in the detection of the tumor as it utilized an ensemble approach that involved multiple pre-trained models. Namely, the CNN and VGG16 models combined reached the accuracy rates of 0.97687 on the validation set and 0.9801 on the test set. Our results confirmed the usefulness and precision of our method to MRI image-based brain tumor classification.

In order to determine which of the eight pre-trained CNN models, initialized with ImageNet weights, was the most successful, Durga et al. [16] suggested assessing the models' performance. On a publicly available MRI dataset, they performed experiments and discovered that the suggested approach performed better than other popular DL frameworks, with the highest accuracy of 96 percent. Some of the evaluation metrics used to measure the performance of the combined model included accuracy, precision, recall, and F1-score. Optimization was done using Adam optimizer. This paper shows that deep learning techniques can be used in medical image processing, especially in helping physicians to detect brain cancer at an early stage.

## Critical review and observed limitations of prior studies

Despite the fact that several deep learning and hybrid models have shown high accuracy in detecting brain tumors including ResNet-based HOG frameworks [9], CNN architecture [10], U-Net segmentation [11], and attention-guided CNNs [12–14], a number of methodological weaknesses still exist. In the majority of current research, numerical performance is prioritized, which is not interpretable or clinically explainable. The datasets provided (e.g., Figshare, Br35H, Harvard) tend to be small, single-centered, and not representative of scanners and patient demographics, which restricts their generalizability. In addition, the previous models only use deep features without considering handcrafted radiomic descriptors, which are able to capture texture heterogeneity needed to grade tumors. Transformer-based and GAN-based architectures are accurate, but computationally expensive and not suitable to be deployed in clinical real-time. These shortcomings highlight the importance of hybrid systems that combine radiomics with deep learning to provide precision and interpretability in brain tumor detection.

## Limitations of existing systems

Conventional CNN-based models and traditional image processing methods tend to falsely identify small or irregular tumors, thus resulting in false positives and negatives. The quality of MRI acquisitions is so important to their performance, and it can be different in equipment and institutions. Numerous deep learning systems are black boxes and need professional interpretation, which makes them prone to change in the results of the diagnosis. Moreover, not many models combine handcrafted radiomics functionality, including GLCM-based texture descriptors, and thus, they cannot describe tumor heterogeneity and fine morphological differences. All these reduce the robustness, interpretability, and cross-dataset consistency of the model, and the proposed solution to these issues is the Mask R-CNN with radiomics integration.

## Research gap

Although great progress has been made in brain tumor detection with DL and image processing methods, problems still remain in terms of high accuracy, early tumor identification, and high-quality classification of various tumor types. A large number of current techniques are unable to extrapolate to novel datasets or be used to detect nuanced tumor characteristics in low-quality images. Also, the combination of classical features, like radiomics, and DL models to improve tumor characterization and classification is a poorly studied area. Such gap leaves a room to advance diagnostic devices to provide increasingly accurate and prompt brain tumor diagnostics.

## Methodology

### Proposed model

In this section, discuss how MRI scans can be applied to enhance brain tumor detection and segmentation models using R-CNN, radiomics and GLCM. Mask R-CNN is an algorithm that estimates tumor regions of interest (ROIs) through pixel-wise segmentation. Out of the segmented ROIs, radiomics features, including texture, shape, and intensity measures, are obtained to provide a quantitative assessment of the heterogeneity within the tumors. Meanwhile, GLCM-based texture features are computed to derive fine-grained spatial relations of pixel intensities in tumor patches which give more information on tumor structure. Deep features acquired in the Mask R-CNN architecture are paired with such handcrafted features to create a strong set of features that has the advantages of classical radiomics and deep learning. The joint feature set is then used along with an MLP classifier to identify the presence and the type of tumor. Fig 1 demonstrates the Mask R-CNN-GLCM block diagram.

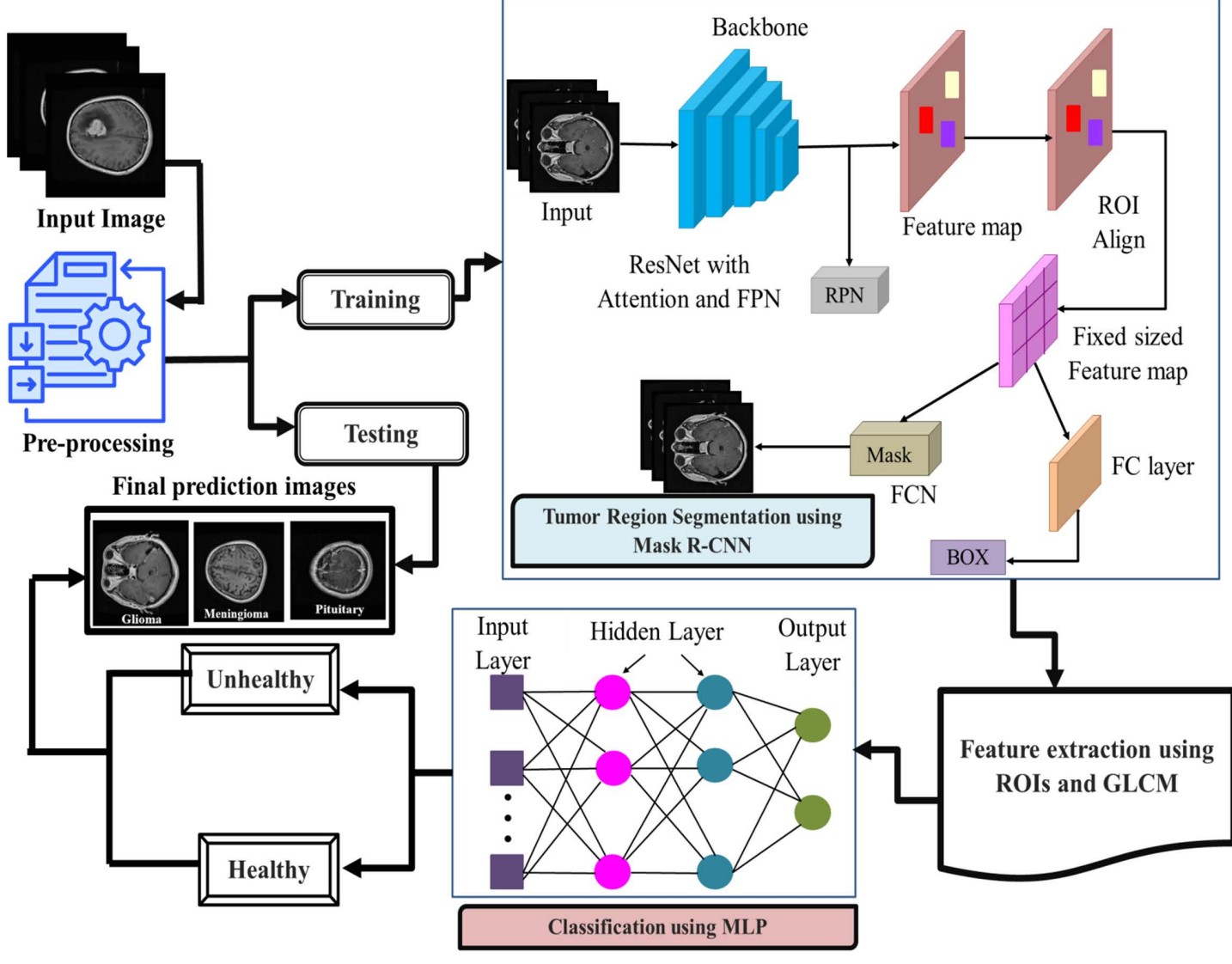

**Fig 1. Block diagram of the Mask R-CNN- GLCM.**

## Dataset

This work employed two publicly available brain tumor MRI datasets: the Harvard Whole Brain Atlas [17] and the Figshare Brain Tumor Dataset [18]. While the Harvard Whole Brain Atlas was solely utilized for external validation and visualization, the Figshare dataset was the main source for model training and testing.

The primary dataset used in this study is the Figshare Brain Tumor MRI Dataset. It consists of 3,064 T1-weighted contrast-enhanced MRI scans, categorized into Glioma (1,426 images), Meningioma (708 images), and Pituitary Tumor (930 images). The images are saved in PNG format and 512 x 512 pixels (grayscale). To develop the model, stratified random sampling was used to divide the dataset into training (80 percent, 2,451 images), validation (10 percent, 307 images), and testing (10 percent, 306 images) subsets to have equal representation of tumor classes.

The external validation data was taken as the Harvard Whole Brain Atlas that provides expert-labeled MRI images of normal and pathological brain conditions, such as glioma, meningioma, and others. This data was applied to qualitative

validation only to determine the accuracy of segmentation and visualize tumor morphology. Fig 2 shows the figshare dataset. Fig 3 illustrates tumor class distribution in Figshare dataset.

## Preprocessing techniques

In order to achieve reproducibility and consistency among experiments, a set of pre-processing methods were used systematically before segmentation and feature extraction. Such methods are normalization, data augmentation and noise reduction.

## Normalization

Normalization was also used to standardize all MRI image intensities to a standard scale, which reduced the differences that would occur due to scanner differences or acquisition parameters. Equation (1) shows how the value of each pixel intensity I was rescaled with the help of minmax normalization to the range [0,1]:

$$I' = \frac{I - I_{min}}{I_{max} - I_{min}}$$

(1)

This is a step to make sure the deep learning model is sensitive to the significant structural patterns and not random differences in intensities, which increases the stability of convergence and reproducibility of the model across datasets.

## Data augmentation

Augmentation was used to the training set in order to solve data imbalance and enhance generalization of the model. The transformations that were introduced in a systematic manner included the following:

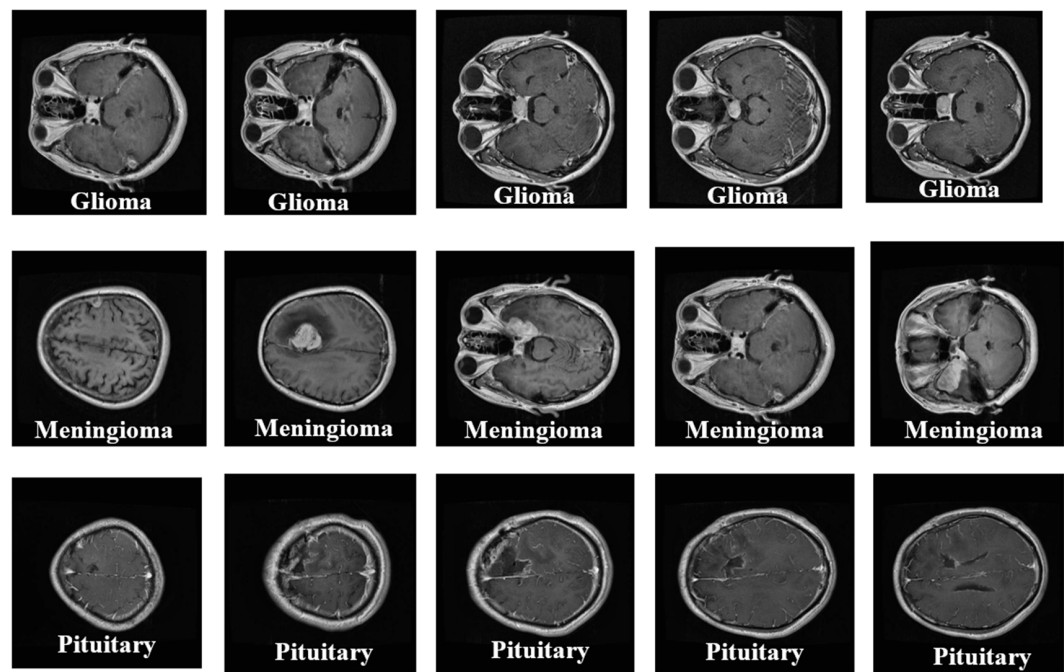

**Fig 2. Figshare dataset.**

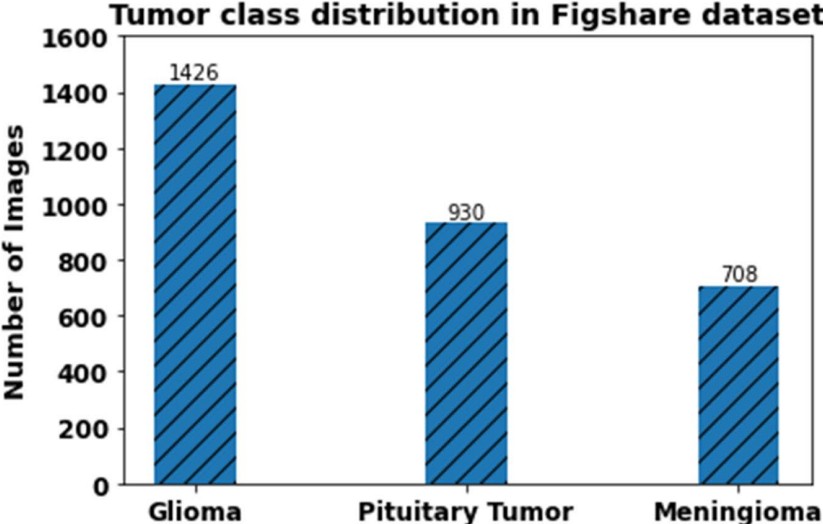

**Fig 3. Tumor class distribution in Figshare dataset.**

- Rotation: Random rotations within ±15° are applied to account for variations in head positioning.

- Flipping: Horizontal and vertical flips are applied to improve invariance to orientation.

- Zooming and Translation: Minor random zoom (±10%) and movements are introduced to create the effect of image framing variability.

- Elastic Deformation: Applied to mimic realistic tissue deformation.

  All transformations were done using controlled random seeds so that the experiments could be perfectly reproduced.

## Noise reduction

MRI images often contain Gaussian or Rician noise that can obscure fine details. A 2D Gaussian filter (GF) was applied with a kernel size of 5×5 and a standard deviation of σ = 1.0. This filter smooths intensity variations while preserving important tumor boundaries, as demonstrated in Fig 4 of the manuscript.

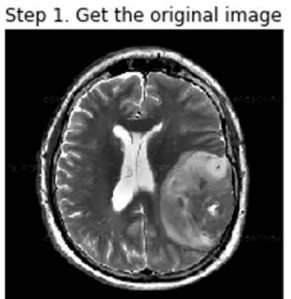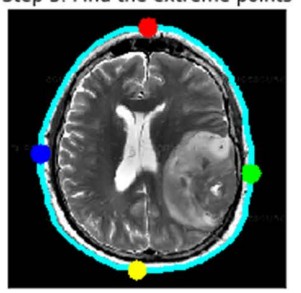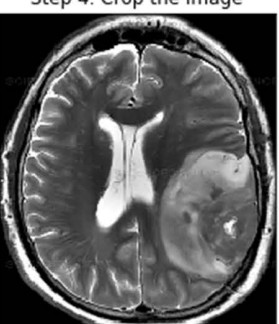

**Fig 4. Pre-processing images of brain tumor.**

To enhance the samples and eliminate noise, we have employed a 2-dimensional GF. Nevertheless, it requires more processing power, which opens a new field of research. At this stage, the convolutional operators are modeled as the Gaussian operators, in which the convolution is used to mean smoothness [19]. The following equation (2) illustrates the 1-D Gaussian operator:

$$G_{1d}^{(a)} = \frac{1}{\sqrt{2\pi\partial}} e^{-\left(\frac{a^2}{2\partial^2}\right)}$$

(2)

Localization in the domains of space and frequency is not affected by the significant filter for sample smoothness; however, the uncertainty function is implemented as shown in equation (3):

$$\Delta a \Delta w \geq \frac{1}{2}$$

(3)

The 2-D Gaussian operator is shown below, as indicated in equation (4):

$$G_{2d}(a, b) = \frac{1}{2\pi\partial^2} e^{-\left(\frac{a^2+b^2}{2\partial^2}\right)}$$

(4)

GF's standard deviation is shown by sigma ($\partial$). If the value is maximum, the smoothness will be maximum as well. In contrast, the sample displaying the window magnitudes' cartesian coordinates are shown in ($a, b$).

## Tumor Region Segmentation using Mask R-CNN

Mask R-CNN is an extension of Faster R-CNN intended for instance segmentation. In segmentation tasks, objects in an image are identified and categorized, while a pixel-wise mask is generated for each object [20]. It is especially applicable in medical imaging work, e.g., brain tumor diagnosis, when it is essential to identify the tumor and outline its shape and boundaries with precision.

## Improvement over standard CNN architectures

In contrast with the traditional CNN models, which only classify the images on an image-level basis, Mask R-CNN (a variant of R2CNN) is proposed to detect the possible tumor-containing regions in the images, which are further classified by RoI Align to provide a more accurate spatial alignment. This two-step method helps a great deal to increase the accuracy of localization, because the model is trained to concentrate on tumor-related subregions instead of full slices of MRI. Moreover, with a mask prediction branch added, pixel-wise tumor segmentation is also possible, where the irregularly shaped lesions are outlined precisely. These improvements are able to overcome the rough localization constraints of conventional CNNs, which allow precise delineation of tumor regions even in complicated or noisy MRI scans. In Fig 5, the Mask R-CNN's design is shown. Two steps make up the Mask R-CNN model's operation:

1. Region Proposal Network (RPN): The result of this component of the model is a list of potential locations (proposals) of objects that could be in the picture. These regions are then categorized as region proposals and thereafter, they are either determined to have a brain tumor or not.

2. RoI Align & Mask Prediction: On a region proposal, Region of Interest (RoI) features are obtained by applying a process known as RoI Align that is capable of giving better spatial accuracy. The second stage uses these features to make predictions about the class of the object and a pixel-wise mask which shows the exact shape of the tumor.

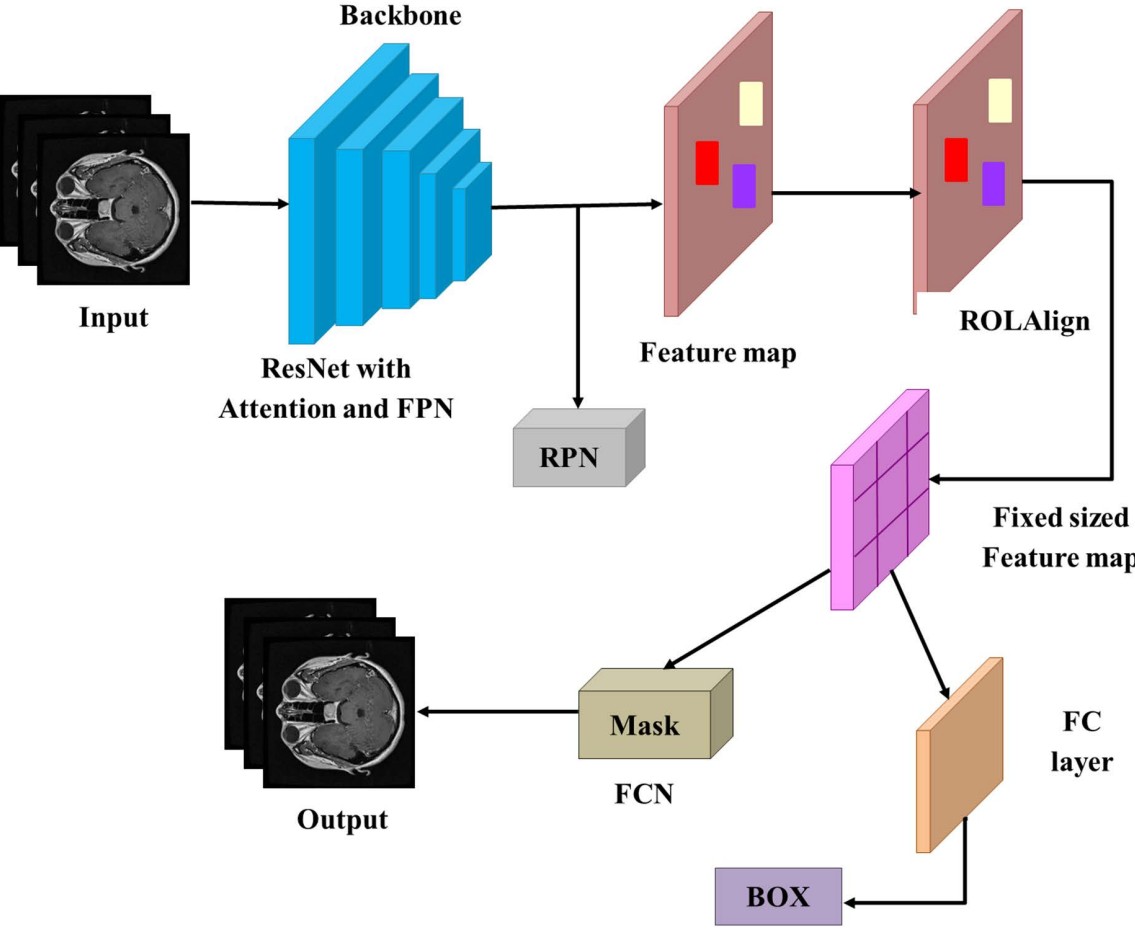

**Fig 5. Architecture of the Mask R-CNN.**

**Region Proposal Network (RPN):** Production of the candidate object proposals with respect to the image is the role of the RPN. It gives an estimation of an objectness score and bounding box predictions of each anchor as illustrated below in equation (5).

$$\hat{y}_i = sigmoid\,(s_i) \tag{5}$$

Where $\hat{y}_i$ is the predicted likelihood that anchor $i$ contains an objective (e.g., tumor), and $s_i$ is the score predicted by the RPN for anchor $i$.

**Bounding Box Regression:** For each proposed objective, the bounding box is refined to better localize the tumor in the image. The regression equation is represented as shown in Equation (6):

$$\hat{b}_i = b_i + \delta_i \tag{6}$$

Where $b_i$ is the ground truth bounding box, and $\delta_i$ is the forecast offset that adjusts the anchor box to the ground truth.

**Mask Prediction:** For each detected region of interest (RoI), a binary mask is generated to classify each pixel in the region as either part of the tumor or not. This mask prediction is modeled using a fully convolutional network (FCN) for each proposal, as illustrated below in equation (7).

$$M_i = FCN(RoI_i) \tag{7}$$

Where $M_i$ indicates the predicted binary mask for the $i$-th $RoI_i$ is the region proposal.

**Loss Function:** Mask loss, boundary box regression loss, and classification loss are among the various losses included in the Mask R-CNN model, as illustrated below in equation (8).

$$L = L_{cls} + L_{bbox} + L_{mask} \tag{8}$$

Where $L_{cls}$ the classification loss (cross-entropy) is, $L_{bbox}$ is the boundary box regression loss (smooth L1 loss), $L_{mask}$ is the mask loss (binary cross-entropy).

As demonstrated in Equation (9), the total loss function integrates all three components to ensure accurate brain tumor identification and segmentation.

$$L_{cls} = -\sum_{i=1}^{N} y_i \log(\hat{y}_i) + (1 - y_i) \log(1 - \hat{y}_i) \tag{9}$$

Where $y_i$ is the ground truth label and $\hat{y}_i$ is the forecast probability.

This equation (10) defines the Smooth L1 loss, which penalizes large localization errors and ensures stable gradient updates for bounding box refinement.

$$L_{bbox} = \sum_{i=1}^{N} smooth_{L1}\left(b_i, \hat{b}_i\right) \tag{10}$$

Where $smooth_{L1}$ is the smooth $L1$ loss function used for bounding box regression.

This equation (11) represents the mask loss, which measures pixel-level differences between the predicted and ground-truth masks to ensure accurate tumor segmentation.

$$L_{mask} = -\sum_{p \in mask\ pixels} y_p \log(\hat{y}_p) + (1 - y_p) \log(1 - \hat{y}_p) \tag{11}$$

Wherever $y_p$ and $\hat{y}_p$ are the ground truth and forecast mask pixel values, respectively.

### Feature extraction using ROIs and GLCM

This method of deriving various quantitative features of medical imaging systems, such as MRI, CT, or PET scans, to provide information that may not be apparent, is referred to as radiomics. One of the fundamental methods that examine the ROIs of medical imaging, which is likely to harbor tumors, is called radiomics. As some information is extracted in these ROIs, the nature of tumors can be better understood and this may be applied in the diagnosis, treatment and prognosis planning [20].

In radiomics, a statistical method for measuring texture features in pictures is called the GLCM. In an image, it determines the frequency of pixel pairs with particular spatial connections and intensity values. GLCM helps describe the spatial relationships of pixels in terms of their intensity or color and is widely used to identify patterns, such as tumor heterogeneity in brain scans.

The radiomics, including the ROI-based texture analysis and the GLCM, can be rather useful in the brain tumor detection to increase the accuracy of the tumor classification.

**GLCM Calculation:** GLCM is computed through the spatial analysis of pixel intensities in a ROI. For a given image with pixel values $I(i,j)$ the co-occurrence matrix $P(i,j,d,\theta)$ for a specific direction $\theta$ and distance $d$ is definite as equation (12):

$$P(i, j, d, \theta) = \sum_{x-1}^{M} \sum_{y-1}^{N} \delta(I(x, y) - i, I(x + d\cos(\theta), y + d\sin(\theta)) - j) \tag{12}$$

The image dimensions are represented by $M$ and $N$, the distance between pixel pairs is indicated by $d$, the orientation angle is indicated by $\theta$, which is usually set to 0°, 45°, 90°, or 135°, and the Kronecker delta function is indicated by $\delta(\cdot)$ which returns 1 when the provided condition is true and 0 otherwise.

**GLCM Features (Texture Features):** Several features can be extracted from the GLCM to describe texture patterns, including the following:

- **Contrast:** Measures the intensity contrast between neighboring pixels, which is defined by equation (13):

$$Contrast = \sum_{i,j} (i-j)^2 \cdot P(i,j) \tag{13}$$

- **Correlation:** Evaluates the correlation of a pixel with its neighbors. It is given by equation (14):

$$Correlation = \frac{\sum_{i,j} (i - \mu_i)(j - \mu_j) P(i,j)}{\sigma_i \sigma_j} \tag{14}$$

Where $\mu_i$ and $\mu_j$ are the means of the row and column of the GLCM, correspondingly, $\sigma_i$ and $\sigma_j$ are the regular deviances of the row and column of the GLCM correspondingly.

**Energy (Angular Second Moment):** Evaluates the uniformity of the image, which is defined by equation (15):

$$Energy = \sum_{i,j} P(i,j)^2 \tag{15}$$

**Homogeneity:** Evaluates the relationship between the GLCM diagonal and the element distribution, as expressed in equation (16):

$$Homogeneity = \sum_{i,j} \frac{P(i,j)}{1 + (i-j)^2} \tag{16}$$

**Radiomic Feature Extraction from ROIs:** For the specific ROI containing the brain tumor, the following general radiomic features can be extracted from equation (17):

$$Feature_{i,j} = f(P(i, j, d, \theta)) \tag{17}$$

Where $f$ is a function representing any of the texture features (homogeneity, contrast, correlation, energy, etc.) derived from the GLCM. The ROI typically encompasses the tumor, and multiple ROIs can be analyzed to assess tumor heterogeneity.

**Radiomics Model for Brain Tumor Detection:** These extracted features can then be input into ML models for classification. The classification can be represented by equation (18):

$$y = classifier(Features\,from\,GLCM\,and\,ROIs) \tag{18}$$

Where *y* is the predicted class (e.g., benign or malignant). The classifier is trained on a labeled dataset using the extracted texture features.

## Feature fusion mechanism

To integrate the handcrafted radiomics and GLCM features with the deep features from the Mask R-CNN backbone, a feature-level concatenation strategy was applied. The deep feature vector extracted from the Mask R-CNN (dimension $D_1$) and the handcrafted feature vector comprising radiomics and GLCM descriptors (dimension $D_2$) were concatenated to form a unified representation of dimension $D_1 + D_2$. Before fusion, both vectors were normalized using z-score standardization to ensure balanced feature scaling. The combined vector was then passed to the MLP classifier for tumor classification. This concatenation approach effectively preserves both low-level spatial semantics from deep learning and high-level handcrafted texture cues from radiomics. In future extensions, adaptive fusion strategies, such as attention weighting or dimensionality-reduction–based projection (e.g., PCA or autoencoder embedding), may be explored to further enhance feature complementarity and reduce redundancy.

## Detection head (MLP) using fused features

Multiple layers of nodes, or neurons, comprise an ANN type called a MLP, with each layer fully connected to the one behind it [21–22]. This is a feedforward neural network structure that is normally applied in both classification and regression. In medical imaging, an MLP can be used to identify brain tumours by categorising MRI data as either tumour present or tumour absent. Fig 6 establishes the architecture of the MLP.

The MLP model typically consists of:

**Input Layer:** The input layer receives intensities of the pixels that are received in the MRI images or the qualities of the pixels that are received in the MRI images. Hidden layers consist of layers of neurons that learn complex features by making non-linear transformations. The output layer gives the last categorization (e.g., 1 tumor found, 0 no tumor).

The MLP has several layers that transform their inputs with the help of weights, biases, and activation functions. No matter the nature of the problem, output is usually subjected to a non-linear activation function like softmax, ReLU or sigmoid.

The following equation (19) regulates the output of a neuron in a specific layer:

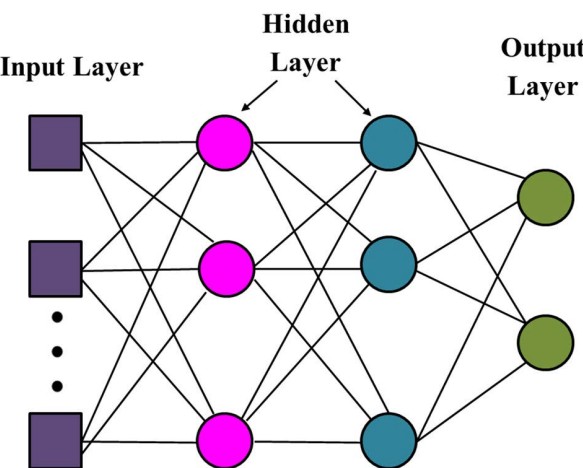

**Fig 6. MLP Architecture diagram.**

$$z^{(l)} = W^{(l)}x^{(l-1)} + b^{(l)} \tag{19}$$

Wherever $z^{(l)}$ is the input to the neurons in the $l$-th layer, $W^{(l)}$ is the weight matrix of the $l$-th layer, $x^{(l-1)}$ is the input from the previous layer $x^{(0)}$ being the raw features or image data), $b^{(l)}$ is the bias term for the $l$-th layer.

The output for the neuron in layer $l$ is system by applying an activation function $\sigma(\cdot)$ is illustrated below, as displayed in equation (20):

$$a^{(l)} = \sigma\left(z^{(l)}\right) \tag{20}$$

In this case, $a^{(l)}$ stands for layer $l$ neuron activity.

Equation (21) provides the final output in the binary classification output layer:

$$\hat{y} = \sigma\left(W^{(L)}a^{(L-1)} + b^{(L)}\right) \tag{21}$$

Wherever $\hat{y}$ is the predicted probability of the tumor being present (using a sigmoid function for binary organization), $L$ represents the final layer of the MLP.

## Loss function and optimization

Equation (22) below shows how the system is trained to minimize the loss function, usually using cross-entropy for binary classification:

$$L = -\left[y\log(\hat{y}) + (1-y)\log(1-\hat{y})\right] \tag{22}$$

In this case, $\hat{y}$ stands for the expected probability of tumor presence, and $y$ for the true label.

Gradient descent, or one of its variations such as Adam, is commonly used to optimize an MLP. Equation (23) shows the gradient descent update rule:

$$W^{(l)} \leftarrow W^{(l)} - \eta\frac{\partial L}{\partial W^{(l)}} \tag{23}$$

Where $\eta$ is the learning rate.

## Training process

The training process consists of the following stages:

Forward Pass: Use the layers to calculate the outputs from the input. Determine the loss between the expected results and the true labels. Backward Pass: Backpropagate the error using the gradient descent approach to update the weights and biases.

## Advantages of proposed method

- Radiomics and GLCM allow obtaining quantitative and detailed texture features of medical images. The combination enhances the capacity of the model to detect subtle tumor features, including heterogeneity, which cannot otherwise be identified in the event of the utilization of conventional imaging methodologies.

- Moving forward, the MLP can utilize the features of GLCM and radiomics in the model, which provides an advanced classifier with the ability to learn on high-dimensional features that are classically difficult. The MLP can discern intricate patterns in the data, leading to enhanced difference among tumor kinds or among malignant and benign tumors.

## Result and discussion

### Environmental setup

It was done in a computer with hard disk memory of 73 GB, graphics card of 16GB, RAM of 13GB and two core Intel Xeon processor. Jupyter Notebook was used to perform the experiment. The suggested strategy was developed in Python and common packages, such as Pandas, Matplotlib, Scikit-learn, Keras, TensorFlow, Seaborn, and NumPy.

### Performance metrics

Using CPU is an indicator of the proportion of time used in activities which involve active processing by the CPU, and not idle time. It gives a hint on the efficiency of the CPU usage. Equation (24) computes the proportion of the time spent at the CPU processing tasks out of the total time available.

$$CPU\ Utilization = \left( \frac{CPU\ Active\ Time}{Total\ Observation\ Time} \right)$$

(24)

Peak Signal-to-Noise Ratio (PSNR) is used to determine the quality of a compressed or reconstructed image relative to the original. It compares the differences between pixels to determine how similar they are, normally through Mean Squared Error (MSE). The larger the PSNR values, the better the images. This equation (25) is a comparison of the quality of reconstruction of an image to the original, and is often employed to measure the performance of image compression or de-noising.

$$PSNR = 10 \cdot \log_{10} \left( \frac{MAX^2}{MSE} \right)$$

(25)

Localization accuracy refers to the capability of a system or procedure to accurately determine the position or location of an object, entity, or signal in space. It measures the proximity of the estimated location to the true location. Equation (26) quantifies how closely the detected tumor region matches the true tumor location in spatial terms.

$$Localization\ Accuracy = 1 - \frac{E_d}{d_{max}}$$

(26)

### Comparative methods

**CNN-LSTM [23]:** To prepare the MRI images for the classification stage, conventional computer vision algorithms were employed. Consequently, the DL model introduced for classification was used, along with traditional techniques for preprocessing and feature extraction.

**Caps-VGGNet [24]:** By adding VGGNet layers to the CapsNet framework, this study presents the Caps-VGGNet hybrid architecture, which combines Capsule Network (CapsNet) with VGGNet. By automatically extracting and classifying features, the proposed technology successfully addresses the challenge of requiring large datasets.

### Localization Advantage of Mask R-CNN (R2CNN)

The Mask R-CNN architecture is more accurate in localization (98.98%), because it has a region-proposal-based learning and RoI alignment scheme, which allows it to maintain spatial accuracy unlike the global receptive fields of the conventional CNNs. The mask generation arm of the model offers pixel-level localization, or, in simpler terms, isolating the tumor, unlike the conventional CNNs, which generate rough bounding boxes. Such localization is important in clinical MRI

interpretation because the slightest variation in tumor margins may influence surgical and treatment planning.The comparison in Table 1 and Fig 7 assesses the performance of three approaches—CNN-LSTM, Caps-VGGNet, and the proposed technique across three metrics: CPU Utilization, PSNR (Peak Signal-to-Noise Ratio), and Localization Accuracy, for two classes: Non-Tumor (class 0) and Tumor (class 1). Compared to CNN-LSTM and Caps-VGGNet, the proposed technique demonstrates superior performance in terms of PSNR as well as Localization Accuracy in both classes that implies greater quality and accuracy of the image and better localization of tumors. More so, the proposed solution possesses a

**Table 1. Comparison of models across metrics.**

| Class Name | CNN-LSTM | | | Caps-VGGNet | | | Proposed | | |
|---|---|---|---|---|---|---|---|---|---|
| | CPU Utilization | PSNR | Localization Accuracy | CPU Utilization | PSNR | Localization Accuracy | CPU Utilization | PSNR | Localization Accuracy |
| 0 (Non-Tumor) | 76.45 | 91.11 | 90.34 | 74.81 | 91.34 | 94.55 | 91.88 | 96.55 | 97.33 |
| 1 (Tumor) | 81.45 | 88.13 | 92.33 | 89.45 | 84.31 | 96.15 | 94.98 | 97.71 | 98.98 |

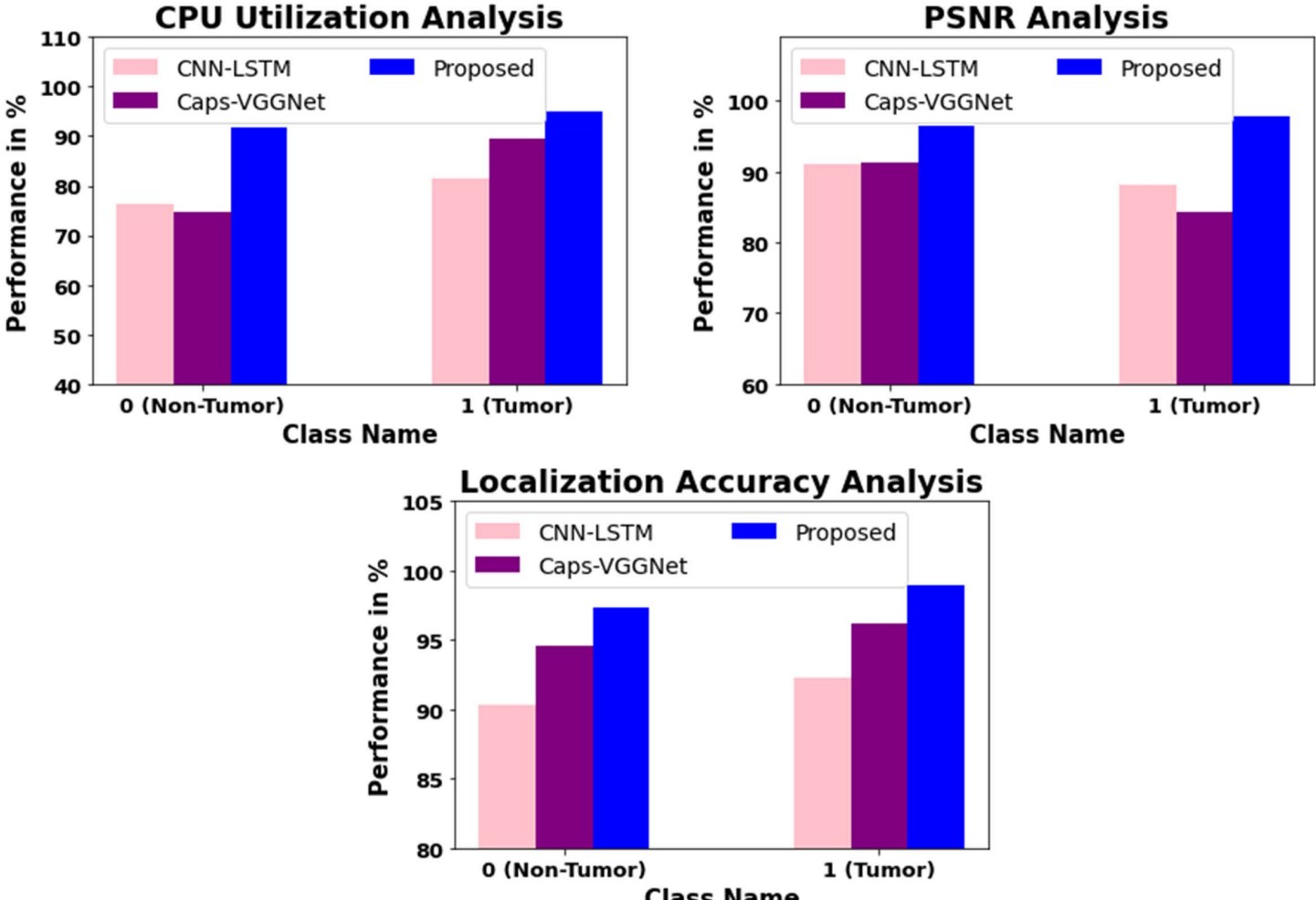

**Fig 7. Comparison of models across metrics.**

slightly high CPU consumption, though it is offset by presumably superior performance indicators, the highest of which are Localization Accuracy (97.33% and 98.98% in non-tumor and tumor cases, respectively).

The impact of processing times in Table 2 and Fig 8 point to the effectiveness of the proposed technique relative to CNN-LSTM and Caps-VGGNet on Non-Tumor (Class 0) and Tumor (Class 1) classifications. In the Non-Tumor case, the proposed method achieves provocatively low processing time of 1.563 seconds, which is superior to CNN-LSTM (6.115 seconds) and Caps-VGGNet (5.334 seconds). The suggested approach defines computational performance that can be used in near-real-time clinical settings. On a workstation with a 2-core Intel Xeon CPU, 13 GB RAM, and a 16 GB GPU, inference required approximately 1.563 seconds for non-tumor and 3.175 seconds for tumor cases. These findings highlight the model's ability to deliver speedy and accurate predictions in time-critical medical imaging workflows. Although radiomics extraction and Mask R-CNN contribute to some computational overhead, the overall processing time remains significantly lower than comparable CNN-LSTM and Caps-VGGNet methods, confirming its feasibility for clinical deployment. Further optimization through model pruning or quantization could enhance performance for strict real-time or intra-operative use.

Similarly, for the Tumor class, the proposed technique demonstrates competitive performance with a processing time of 3.175 seconds, which is faster than CNN-LSTM's 10.876 seconds and close to Caps-VGGNet's 4.114 seconds. These results underscore the model's capability to deliver faster predictions while maintaining superior accuracy and quality, making it highly suitable for time-critical applications in medical imaging.

**Table 2. Processing time analysis for proposed method with existing system.**

| Class Name | CNN-LSTM | Caps-VGGNet | Proposed |
|---|---|---|---|
| | Processing Time | | |
| 0 (Non-Tumor) | 6.115 | 5.334 | 1.563 |
| 1 (Tumor) | 10.876 | 4.114 | 3.175 |

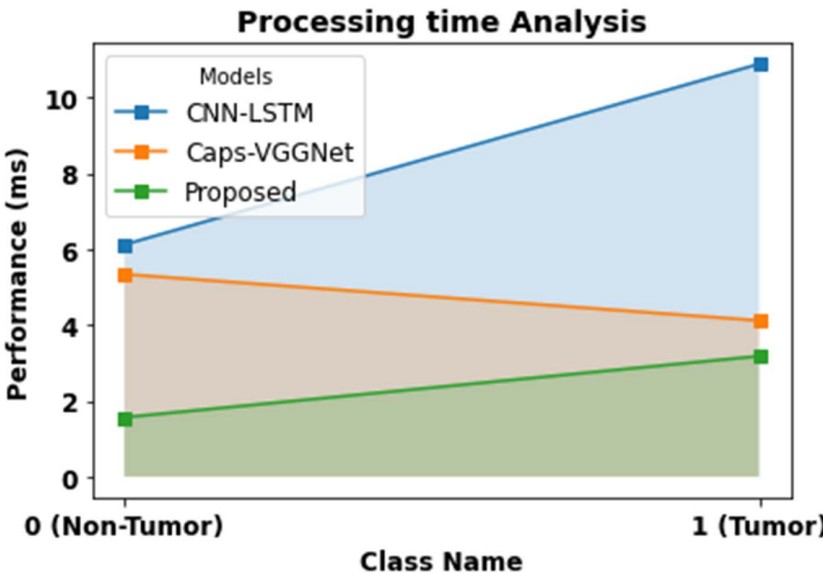

**Fig 8. Processing time analysis for proposed method.**

### Training validation loss and accuracy analysis

Fig 9 shows how DL models are used to detect brain tumors. Accuracy is tracked together with training and validation loss to evaluate the performance of the model. Training loss indicates how successfully the model is trained using the training data, whereas validation loss shows how well the model generalizes to fresh data. While validation accuracy assesses the model's performance on a separate validation set to identify overfitting, training accuracy assesses the model's performance on the training set. For the accurate diagnosis of brain tumors, both the training and validation sets exhibit balanced increases in accuracy and decreases in loss, indicating good model generalization and efficient learning.

### Ablation study

For the ablation study table and graph, we will break down the key components of the system to evaluate their individual contributions to overall performance. The research will analyze the effects of using different combinations of Mask R-CNN, radiomics integration, and GLCM features on the classification accuracy of brain tumor diagnosis. Fig 10 shows the ablation study of accuracy in brain tumor identification methods.

### Role of radiomics features in enhancing detection accuracy

Radiomics features significantly contribute to the improved accuracy of the proposed Mask R-CNN–GLCM framework by providing quantitative descriptors of tumor heterogeneity, texture, and intensity variations that are not visually perceivable in MRI images. These features quantify the internal structure and spatial organization of tumors, enabling a deeper understanding of tissue characteristics and malignancy potential. In particular, GLCM-based texture features including contrast, correlation, energy, and homogeneity analyze spatial relationships between pixel intensities within tumor regions, capturing fine-grained textural irregularities. When these handcrafted features are integrated with deep features extracted by Mask R-CNN, the hybrid representation becomes more discriminative and robust. This synergy improves tumor localization and classification accuracy, as evidenced by superior results (98.15% accuracy and 98.98% localization precision) compared with CNN-LSTM and Caps-VGGNet baselines. Moreover, radiomics enhances the interpretability of the system: its features correlate with histopathological properties, allowing clinicians to relate model outputs to biological meaning. Thus, radiomics in deep learning pipeline can provide quantitative accuracy and clinical transparency to make believable diagnosis of brain-tumors.

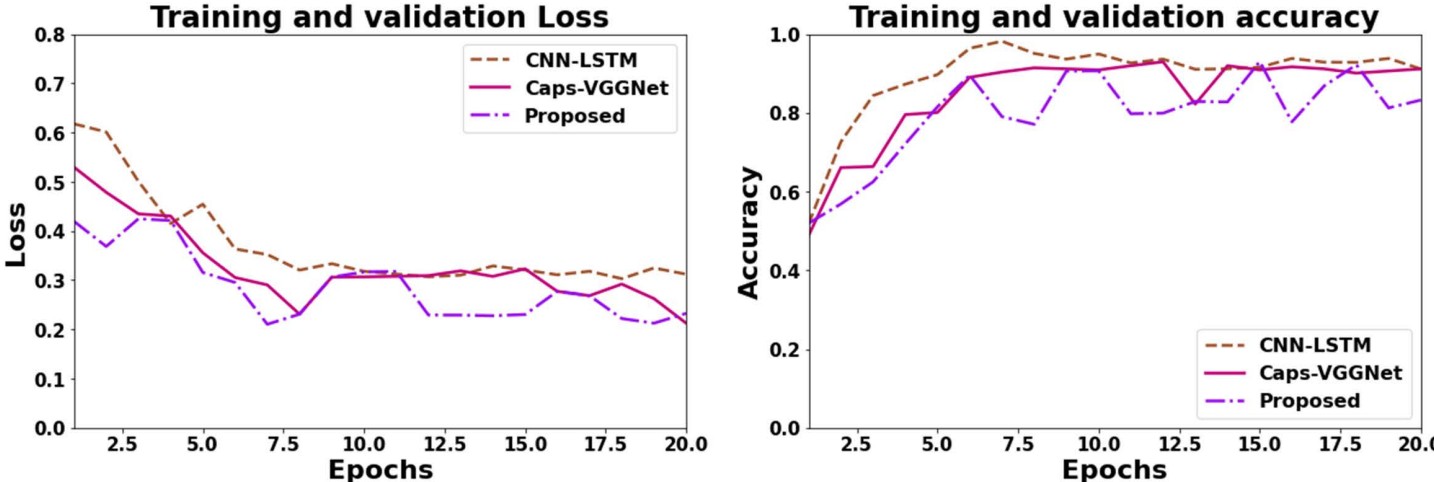

**Fig 9. Training validation loss and accuracy analysis.**

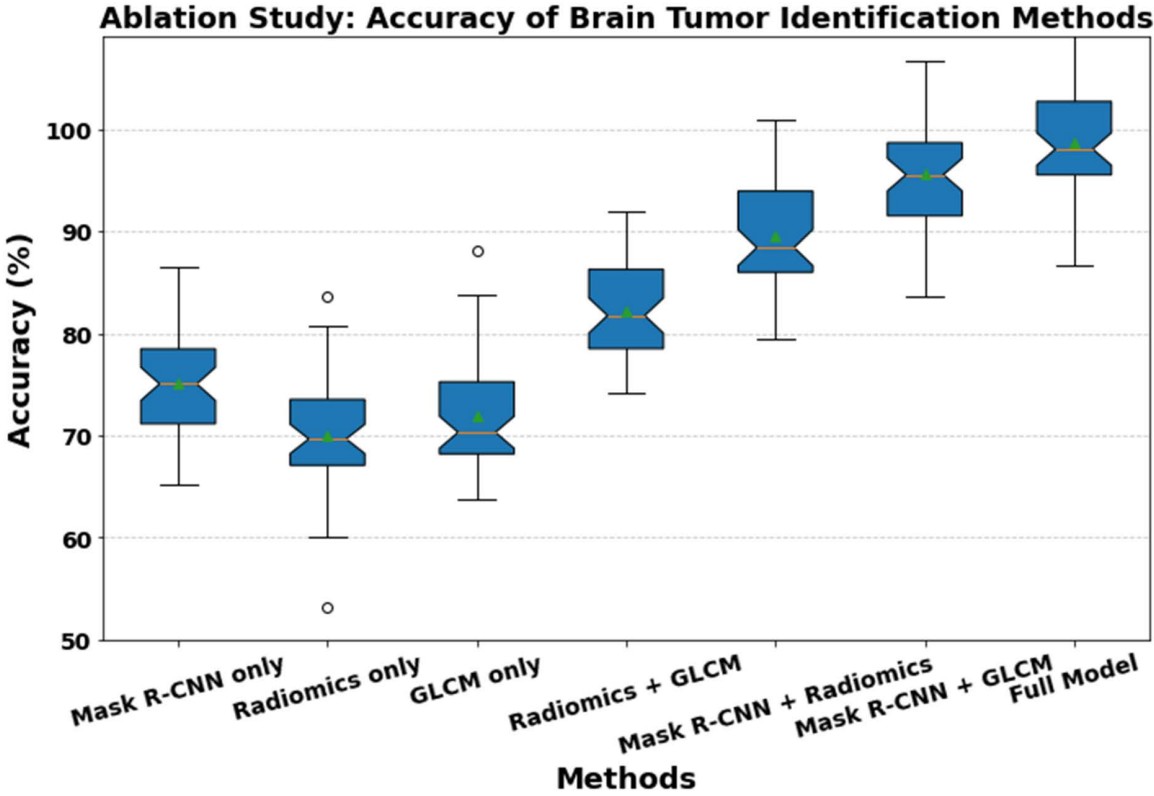

**Fig 10. Ablation study of accuracy brain tumor identification methods.**

## Influence of GLCM

The GLCM can accomplish the analysis of the texture property through the spatial relationship measure of pixel intensities in an image. It measures the trend of contrast, correlation and homogeneity of the tumor area and gives useful information regarding the inner structure and heterogeneity of the tumor. GLCM recovers the performance of brain tumor detection models with rich and complementary features that can further classify and characterize tumors and is an indispensable aspect when applied alongside DL models such as Mask R-CNN and radiomics.

Table 3 and Fig 11, which compare brain tumor detection approaches, demonstrate how deep learning techniques have advanced. A refined Inception-V3 model was applied to brain MRI data by Noreen et al. with an accuracy of 94.71%. MobileNetV2, as used by Asiri et al., has a little lower accuracy of 92.11%. Using a refined Deep-Net model, Khan et al. achieved an improved accuracy of 95.61%. With an accuracy of 98.15%, the combination of Mask R-CNN and GLCM in

**Table 3. The proposed method was compared to other methods.**

| Author | Methodology | Dataset | Accuracy |
|---|---|---|---|
| **Noreen et al. [25]** | Inception-V3Fine-tunedmodel | BrainMRI | 94.71 |
| **Asiri et al. [26]** | MobileNetV2 | BrainMRI | 92.11 |
| **Khan et al. [27]** | Deep-Net:Fine-Tunedmodel | BrainMRI | 95.61 |
| **Our Model** | Mask R-CNN- GLCM | Figshare dataset | 98.15 |

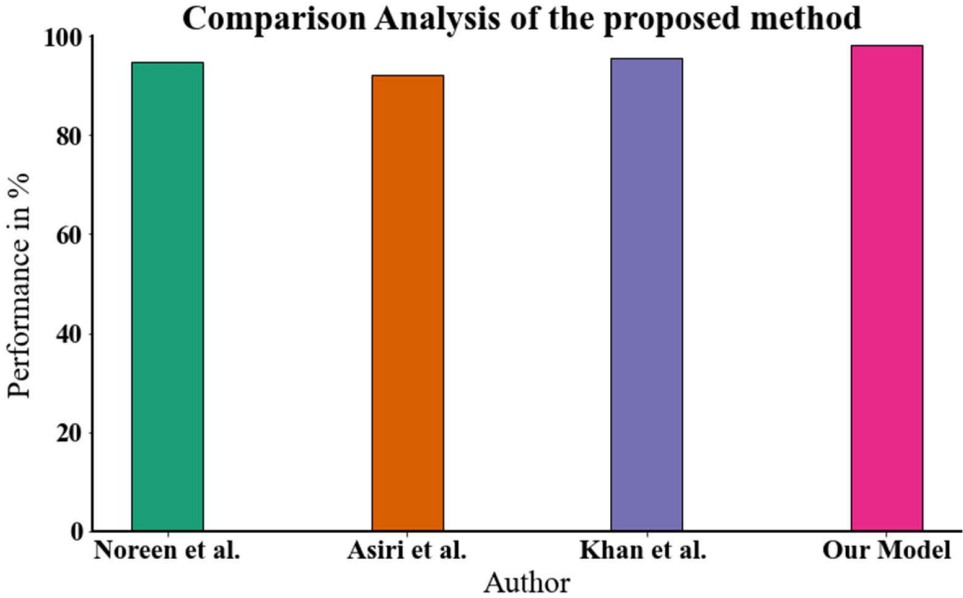

**Fig 11. The proposed method was compared to other methods.**

our suggested method on the Figshare data set demonstrated superior efficacy, demonstrating the possible of both DL and radiomics in the detection of brain cancers.

## Challenges and limitations

The integration of Mask R-CNN with radiomics, GLCM, and MLP for brain tumor detection presents numerous challenges and limitations. One primary challenge is the computational complexity, as processing radiomics features and integrating them with deep learning models like Mask R-CNN requires significant computational resources. Additionally, the variability in brain tumor types and sizes may result in difficulties when generalizing the model across different datasets. Furthermore, accurately extracting and selecting relevant GLCM features for tumor characterization is complex and requires careful optimization.

## Conclusion

This research describes a novel and advanced brain tumor detection and classification system that applies both Mask R-CNN, Radiomics features namely the GLCM, and a detection head (MLP). Combination of the quantitative textual features of radiomics with the deep contextual learning power of Mask R-CNN would allow the proposed system to effectively systematize the spatial and morphological heterogeneity of tumors in MRI images. The hand crafted and deep features combine to give a synergistic effect which enables localization of the tumor and powerful classification. The experimental results of the Figshare Brain Tumor Dataset prove that the proposed scheme reached the classification accuracy of 98.15% higher than the existing benchmark models of CNN-LSTM and Caps-VGGNet in the localization precision, PSNR, and processing efficiency. The model has additionally achieved immense advancements in the utilization of the CPU and inference time and demonstrated that it can be computationally viable and can be made clinically applicable. The experiments of ablation confirm the significance of radiomics and textural characteristics of GLCM in the formation of feature discriminability and better tumor characterization. This study reveals that hybrid AI models that integrate deep learning with interpretable feature extraction could be significant clinically in early diagnosis and treatment planning. To enhance

the adoption and trust in the medical practice, the combination of pixel-level segmentation and quantitative radiomics should be explainable. Besides, the framework presents the possibility of flexibility of the further medical imaging tasks, which deal with the hard tissue textures, e.g., the liver lesions or pulmonary nodules.

### Limitations and future work

Although the framework has promising outcomes, there are a number of challenges associated with it. Radiomics extraction and deep learning integration raise the training time and resource demands because of their computational complexity. Moreover, the datasets are restricted to publicly available collections of MRI, which might not be representative of scanner-to-scanner, institution-to-institution, or patient-to-patient variation. Therefore, the extrapolation to multicentre or real world data is one of the major avenues of future study.

Future works include optimization of computational efficiency by compressing models and training them on multiple machines, multi-modal data (e.g., CT, PET, and histopathological images), and diversifying data (e.g., multi-center cohorts). Additionally, integrating explainable AI modules and domain adaptation techniques will further enhance the model's interpretability and cross-domain robustness for clinical translation.

### Author contributions

**Conceptualization:** Prathima Devadas.

**Data curation:** Prathima Devadas.

**Formal analysis:** Prathima Devadas.

**Funding acquisition:** Prathima Devadas.

**Investigation:** Prathima Devadas.

**Methodology:** Prathima Devadas.

**Project administration:** Prathima Devadas.

**Resources:** Gandhi Mathivanan.

**Software:** Gandhi Mathivanan.

**Supervision:** Gandhi Mathivanan.

**Validation:** Gandhi Mathivanan.

**Visualization:** Gandhi Mathivanan.

**Writing – original draft:** Gandhi Mathivanan.

**Writing – review & editing:** Gandhi Mathivanan.

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
