## [Decision Letter · Decision Letter 0]

16 Oct 2025

Dear Dr. Devadas,

Thank you for submitting your manuscript to PLOS ONE. After careful consideration, we feel that it has merit but does not fully meet PLOS ONE’s publication criteria as it currently stands. Therefore, we invite you to submit a revised version of the manuscript that addresses the points raised during the review process.

We look forward to receiving your revised manuscript.

Kind regards,

Subramani Neelakandan

Academic Editor

PLOS ONE

Journal Requirements:

3.PLOS requires an ORCID iD for the corresponding author in Editorial Manager on papers submitted after December 6th, 2016. Please ensure that you have an ORCID iD and that it is validated in Editorial Manager. To do this, go to ‘Update my Information’ (in the upper left-hand corner of the main menu), and click on the Fetch/Validate link next to the ORCID field. This will take you to the ORCID site and allow you to create a new iD or authenticate a pre-existing iD in Editorial Manager.

Additional Editor Comments (if provided):

Author presented the research entitled “Mask - Region-based Convolutional Neural Networks (R-CNN) with Radiomics Integration and Gray Level Co-occurrence Matrix (GLCM) for brain tumor detection” well. Here I would like to point out some suggestion to proceed further.

The manuscript should specify dataset sources (e.g., Figshare, Br35H, BraTS), image count, resolution, and data partition ratios (training/testing).

Clarify preprocessing techniques—normalization, augmentation, and noise reduction—to ensure reproducibility.

The paper mentions radiomics and GLCM integration, but the fusion mechanism (concatenation, attention weighting, or dimensionality reduction) is not clearly explained.

Add an ablation study to show the contribution of each component (Mask R-CNN, radiomics, GLCM).

Ensure consistency in terminology—use either “brain tumor classification” or “detection and segmentation” throughout the manuscript.

Add recent literature references (2022–2025) comparing hybrid deep learning and radiomics models for medical imaging.

Include a “Limitations and Future Work” section discussing scalability, dataset diversity, and model generalizability.

Reviewers' comments:

Reviewer's Responses to Questions

**Comments to the Author**

1. Is the manuscript technically sound, and do the data support the conclusions?

Reviewer #1: Yes

2. Has the statistical analysis been performed appropriately and rigorously?

Reviewer #1: Yes

3. Have the authors made all data underlying the findings in their manuscript fully available?

Reviewer #1: Yes

4. Is the manuscript presented in an intelligible fashion and written in standard English?

Reviewer #1: Yes

Reviewer #1: The manuscript proposes a new technique called “Mask - Region-based Convolutional Neural Networks (R-CNN) with Radiomics Integration and Gray Level Co-occurrence Matrix (GLCM) for brain tumor detection”. This is a research project with scientific and practical significance. The manuscript is presented with a reasonable layout and rich simulation results, demonstrating the effectiveness of the proposed method. To improve the quality of the manuscript, the following ideas should be considered for editing.

Review Comments:

1. Need to change the “the offering of the our work are follows” in the introduction section.

2. Abbreviations only need explanation the first time they are used. There are a number of abbreviations that are explained multiple times in the manuscript. For example, CNN is explained twice in the literature.

3. Limited literature review has been carried out, so limitations of existing works are not reflected.

4. How does the R2CNN architecture improve tumor region localization compared to standard CNN models?

5. Many of the equations are part of the related sentences. Attention is needed for correct sentence formation.

6. What role do radiomics features play in enhancing the detection accuracy of brain tumors?

7. The manuscript does not discuss potential limitations such as scalability, real-world applicability, or robustness to unseen data.

8. Is the proposed method computationally efficient enough for real-time or clinical application?

9. Rectify the minor errors observed in the paper through thorough copy editing. Improve the writing style, English language, and grammar.

10. Need to improve the conclusion section.

**Do you want your identity to be public for this peer review?** For information about this choice, including consent withdrawal, please see our Privacy Policy

Reviewer #1: No

---

## [Author Response · Author response to Decision Letter 1]

24 Nov 2025

Additional Editor Comments (if provided):

Author presented the research entitled “Mask - Region-based Convolutional Neural Networks (R-CNN) with Radiomics Integration and Gray Level Co-occurrence Matrix (GLCM) for brain tumor detection” well. Here I would like to point out some suggestion to proceed further.

The manuscript should specify dataset sources (e.g., Figshare, Br35H, BraTS), image count, resolution, and data partition ratios (training/testing).

Response: Thank you for your comment. The manuscript has been updated to clearly specify the dataset sources (Figshare and Harvard Whole Brain Atlas), including image count, resolution, and data partition ratios for training, validation, and testing.

Clarify pre-processing techniques—normalization, augmentation, and noise reduction—to ensure reproducibility.

Response: Thanks for your comment. We have added the pre-processing techniques section, which now includes details on normalization, augmentation, and noise reduction. These steps are critical for ensuring reproducibility across experiments.

The paper mentions radiomics and GLCM integration, but the fusion mechanism (concatenation, attention weighting, or dimensionality reduction) is not clearly explained.

Response: Thanks for your comment. We have added the Feature Fusion Mechanism section

Add an ablation study to show the contribution of each component (Mask R-CNN, radiomics, GLCM).

Ensure consistency in terminology—use either “brain tumor classification” or “detection and segmentation” throughout the manuscript.

Response: Thanks for your comment. We have added an ablation study to demonstrate the contribution of each component—Mask R-CNN, radiomics, and GLCM—to the overall model performance. The corresponding results and discussion have been included in the revised manuscript. Additionally, we have ensured consistency in terminology throughout the manuscript by uniformly using the term “brain tumor detection and segmentation.

Add recent literature references (2022–2025) comparing hybrid deep learning and radiomics models for medical imaging.

Response: Thank you for your insightful comment. Recent 2022–2025 references added.

Include a “Limitations and Future Work” section discussing scalability, dataset diversity, and model generalizability

Response: Thank you for your valuable comment. A section on Limitations and Future Work has been added after the Conclusion. This section addresses the challenges related to scalability, dataset diversity, and the generalizability of the model, and outlines potential directions for future research.

Reviewers' comments:

Reviewer's Responses to Questions

Comments to the Author

1. Is the manuscript technically sound, and do the data support the conclusions?

Reviewer #1: Yes

2. Has the statistical analysis been performed appropriately and rigorously?

Reviewer #1: Yes

3. Have the authors made all data underlying the findings in their manuscript fully available?

Reviewer #1: Yes

4. Is the manuscript presented in an intelligible fashion and written in standard English?

Reviewer #1: Yes

5. Review Comments to the Author

Reviewer #1: The manuscript proposes a new technique called “Mask - Region-based Convolutional Neural Networks (R-CNN) with Radiomics Integration and Gray Level Co-occurrence Matrix (GLCM) for brain tumor detection”. This is a research project with scientific and practical significance. The manuscript is presented with a reasonable layout and rich simulation results, demonstrating the effectiveness of the proposed method. To improve the quality of the manuscript, the following ideas should be considered for editing.

Review Comments:

1. Need to change the “the offering of the our work are follows” in the introduction section.

Response: Thank you for your comment. We have revised the phrase “the offering of our work are follows” to “the contributions of our work are as follows” in the Introduction section.

2. Abbreviations only need explanation the first time they are used. There are a number of abbreviations that are explained multiple times in the manuscript. For example, CNN is explained twice in the literature.

Response: Thank you for your comment. We have carefully reviewed the manuscript and ensured that each abbreviation is defined only at its first occurrence (e.g., CNN, MRI, MLP). Repeated explanations have been removed for consistency and clarity.

3. Limited literature review has been carried out, so limitations of existing works are not reflected.

Response: Thank you for your comment. We have added the Critical Review and Observed Limitations of Prior Studies and corrected limitations of exiting model.

4. How does the R2CNN architecture improve tumor region localization compared to standard CNN models?

Response: Thank you for your comment. We have added the content in section Improvement over Standard CNN Architectures and Localization Advantage of Mask R-CNN (R2CNN).

5. Many of the equations are part of the related sentences. Attention is needed for correct sentence formation.

Response: Thank you for your comment. We have carefully reviewed the manuscript and revised the related sentences to ensure that all equations are correctly integrated within their corresponding textual context for improved readability and grammatical accuracy.

6. What role do radiomics features play in enhancing the detection accuracy of brain tumors?

Response: Thank you for your comment. Radiomics features quantitatively capture tumor texture, shape, and intensity variations, enhancing the model’s ability to identify subtle heterogeneity and improving the accuracy of brain tumor detection.

7. The manuscript does not discuss potential limitations such as scalability, real-world applicability, or robustness to unseen data.

Response: Thank you for your comment. We appreciate this valuable observation. The revised manuscript now includes a dedicated section titled “Limitations and Future Work”, which explicitly discusses constraints related to scalability, real-world deployment, computational complexity, and generalization to unseen datasets. This section also outlines strategies for enhancing robustness and cross-domain applicability in future research

8. Is the proposed method computationally efficient enough for real-time or clinical application?

Response: Thank you for your comment. The proposed method demonstrates computational efficiency suitable for near–real-time clinical application, with inference times of approximately 1.563 s (non-tumor) and 3.175 s (tumor), confirming its feasibility for clinical use. We have added the answer in the Results and Discussion section, following the Processing Time Analysis (Table 2 and Fig. 8) subsection.

9. Rectify the minor errors observed in the paper through thorough copy editing. Improve the writing style, English language, and grammar.

Response: Thank you for your comment. We have corrected the grammatical in the manuscript.

10. Need to improve the conclusion section.

Response: Thank you for your comment. We have improved the conclusion section.

---

## [Decision Letter · Decision Letter 1]

19 Jan 2026

Mask - Region-based Convolutional Neural Networks (R-CNN) with Radiomics Integration and Gray Level Co-occurrence Matrix (GLCM) for brain tumor detection and segmentation

PONE-D-25-34631R1

Dear Dr. Devadas,

We’re pleased to inform you that your manuscript has been judged scientifically suitable for publication and will be formally accepted for publication once it meets all outstanding technical requirements.

Kind regards,

Subramani Neelakandan

Academic Editor

PLOS One

Additional Editor Comments (optional):

Author revised the manuscript based on reviewer comments therefore article will be recommended to consider next process.

Reviewers' comments:

Reviewer's Responses to Questions

**Comments to the Author**

Reviewer #1: All comments have been addressed

2. Is the manuscript technically sound, and do the data support the conclusions?

Reviewer #1: Yes

3. Has the statistical analysis been performed appropriately and rigorously?

Reviewer #1: Yes

4. Have the authors made all data underlying the findings in their manuscript fully available?

Reviewer #1: Yes

5. Is the manuscript presented in an intelligible fashion and written in standard English?

Reviewer #1: Yes

Reviewer #1: (No Response)

**Do you want your identity to be public for this peer review?** For information about this choice, including consent withdrawal, please see our Privacy Policy

Reviewer #1: No

---

## [Editor Report · Acceptance letter]

PONE-D-25-34631R1

PLOS One

Dear Dr. Devadas,

I'm pleased to inform you that your manuscript has been deemed suitable for publication in PLOS One. Congratulations! Your manuscript is now being handed over to our production team.

Kind regards,

on behalf of

Dr. Subramani Neelakandan

Academic Editor

PLOS One